# Essential Patents and Knowledge Position, a Network Analysis on the Basis of Patent Citations

**Jiaming Jiang** [1,*] and **Xingyuan Zhang** [2]

1   Graduate School of Humanities and Social Science, Okayama University, Okayama 700-8530, Japan
2   Faculty of Economics, Okayama University, Okayama 700-8530, Japan; zhxy@okayama-u.ac.jp
*   Correspondence: jiaming@okayama-u.ac.jp

**Abstract:** Technology standards are considered important tools for increasing bargaining power and licensing revenues by combining the strategies of firms with the standard-setting organizations (SSOs) standardization processes. The essential patents declared by members of the SSOs play a critical role in such standardization processes. Some former researchers have found that, when using network analysis for measuring the knowledge positions in the "main-path" of standards-based markets, the essential patents did not match very well with the actual knowledge positions of the firms, in most cases. In this paper, we focus on the essential patents declared by the member firms in JTC1, an SSO that provides a standards development environment related to the development of the worldwide information and communication technology (ICT) standards for business and consumer applications, and that employs social network analysis techniques to investigate the knowledge positions of the patents, not only in the "main-path" discussed in the earlier literature, but also in the brokerage processes. We found that the brokerage-process approach helped us to better understand the roles of the essential patents that dominate transactions, relations, and the exchange of knowledge in the patent citation network than that of the main-path. Our findings suggest that claiming essentiality depends on the strategic behavior not only of the patents' owners, but also of the SSOs.

**Keywords:** standard-essential patents; technology standards; standard-setting organizations; social network analysis





## 1. Introduction

As Bekkers and Updegrove (2012) [1], Baron and Pohlmann (2018) [2], and Baron and Spulber (2018) [3] have indicated, technology standards documents describe, define, and codify technologies. Technology standards define commonly accepted techniques, and reflect an agreement between different individuals, firms, or other entities, to use a particular method, which may be novel or not [2]. Technology standards may also govern access to technology because standard-setting organizations (SSOs) often require their members to license the proprietary technology that is necessary for the implementation of a standard, on specified terms [3].

On the other hand, technology standards can prescribe methods that are protected by patents. If a standard cannot be implemented without infringing on a patent, this patent is called a standard essential patent (SEP) [2]. Patented methods may also be useful, but not essential, for implementing a standard. A patented method is deemed commercially essential if it is considered indispensable for making any product that complies with the standard, or if existing alternative methods are technologically inferior or not accessible on commercially viable terms [1–5].

Some recent literature has focused on the values or knowledge positions of the SEPs, which people believe to be a critical factor in providing a competitive advantage to firms for extending their business [6]. The rationale is that the more essential patents that a firm owns and claims to the SSO, the stronger is its knowledge position [7,8]. However, on the other hand, Baron and Pohlmann (2015) [2] argue that many patented inventions

are made during the process of standard development (e.g., to address a specific need or problem in a standardized technology), but are not included in the standard. This is because many different firms make contributions to the standards under development, and the contributions are subject to votes by SSO members.

In their recent study, Bekkers and Martinelli (2012) [9] indicate that claims of essentiality are the result of the strategic behavior of the patent's owner instead of actual technical relevance. By using a social network analysis on the patent citation network related to the 3G W-CDMA standard, they proposed an alternative indicator, i.e., a "main-path", to investigate the knowledge positions of the SEPs in the patent citation network, where they assumed that the main-path is an accurate description of the importance of the knowledge position in the patent citation network, and that most of the patents on this main-path should be claimed essential to the standard. However, they show that, among all claimed essential patents, only very few are on the main-path. They argue that the bulk of these claimed patents may be technologically unimportant, and that the link between the SEPs and the knowledge positions may be weaker than expected if people measure it by using the main-path analysis technique in the patent citation network.

Social network analysis has recently become a useful analytical tool, along with patent statistics [10,11]. Network-based techniques, such as the "main-path analysis", were pioneered by Hummond and Doreian (1989) [12]. In the recent past, a number of papers employed this approach for mapping technological trajectories [13–15]. Specific algorithms can be used to identify the "main flow of knowledge" within the patent citation network. This main flow of knowledge is a set of connected patents and citations linking the largest number of patents of the network, thereby accumulating the greatest amount of knowledge flowing through the citations. This path represents, therefore, a local and cumulative chain of innovations consistent with the definition of a technological trajectory. Given the success of this approach in understanding the main flow and the development of patented knowledge, it might be promising for providing insight into the knowledge positions of the firms that own those patents. However, as indicated in Bekkers and Martinelli (2012) [9], the granularity of this method might restrict its usability in this context: even if the full network comprises thousands, or even ten thousand patents, the identified main-path of knowledge is often comprised of a few dozen patents, or even less. This "over selective" problem may result in serious limitations and lead to a misunderstanding of the knowledge positions of the SEPs. On the other hand, Gould and Fernandez (1989) [16] propose a knowledge broker typology framework. The advantage of the broker position in a network is that the participants who are positioned as information brokers between groups with different information backgrounds benefit from information flows, have a positive influence on their quantitative and qualitative output, and can even induce competition or conflict between neighbors who are not linked directly.

In this paper, we attempt to investigate the relationship between the patent claimed by its owner to be essential and the knowledge position of the patents in the patent citation network. We focus on the knowledge positions not only in the "main-path" discussed in the earlier literature, but also in the brokerage processes.

We pay attention to essential patents declared by member firms in JTC1, an SSO that provides a standards development environment related to the development of worldwide information and communication technology (ICT) standards for business and consumer applications.

Our paper offers three main contributions. Firstly, we verified the result of the former research [9] from an entirely different standards-based high-tech field, i.e., the JTC1 main-path. Secondly, in addition to the results of the analysis of the "main-path", we provide our results for the patent citation networks with other characteristics, e.g., betweenness centrality, and brokerage roles, etc., and obtain a different finding, suggesting a strong correlation between the SEPs and their brokerage roles. Finally, we also implement regression analyses for the determinants of the strategies of the SSO members related to the declaration of the

SEPs by employing the timing for cooperation and entry into an industry SSO, and the patent portfolios of the SSO members.

The paper is organized as follows: in Section 2, we describe our dataset; in Section 3, we discuss the social network analysis on the patent citation network; in Section 4, we provide the regression results for the determinants of the declaration of the SEPs; and Section 5 concludes.

## 2. Materials and Methods

In order to test whether the above methodologies result in good indicators of knowledge positions, we needed appropriate data for our selected case. The essential patent analyses, obviously, requires a dataset of the essential patents, whereas network-based analyses require a dataset that contains information about the patent citation relations between these patents. Below, we will briefly describe the datasets and methods.

### 2.1. Essential Patents in the JTC1

As data are most constrained for standard-essential patents, we collected our data for standard essential patents from the Disclosed Standard Essential Patents (dSEP) Database, developed by Bekkers et al., (2012) [17]. The database is based on the archives of thirteen major SSOs and provides a full overview of all of the disclosed IPRs at setting organizations worldwide. The dSEP is cleaned and harmonized, and all of the disclosed patents, or patent applications, of the United States Patent and Trademark Office (USPTO), or the European Patent Office (EPO), are matched against the patent identities in the Worldwide Patent Statistical Database (PATSTAT).

As discussed in Section 1, we focus on JTC1, an SSO that provides a standards development environment related to the development of worldwide ICT standards for business and consumer applications. Since its inception in 1987, JTC1 has brought about a number of very successful and relevant ICT standards in the fields of multimedia (e.g., MPEG), IC cards ("smart cards"), ICT security, database query, programming languages, as well as character sets. Our sample includes 1,149 SEPs, declared by 63 JTC1 member firms, during the period from 1990–2010. Since the JTC1 includes more than 400 technology standards, the member firms may declare the same patent to different standards. Thus, we identified 387 patents, of which 276 patents were published in the USPTO, and 111 were published in the EPO. Then, we used "docdb_family_id", a unique code defined by the PATSTAT for identifying patent families, to clean our sample, and we finally obtained 241 standard essential patents published in the USPTO.

### 2.2. Patent Citations

In the dSEP, every essential patent has a unique and universal application identification, named the "appln_id", as defined by the PATSTAT, which allowed us to be able to merge all the SEPs listed in the dSEP with the information for those patents in the PATSTAT. We then utilized the information of the patent citations for the USPTO patents in the latter database.

Patent citations presumably convey information or knowledge flows between innovations or patent holders. As shown in Figure 1, we concentrated our sample on the patent citation relationships between the JTC1 member firms' patents. We also included patents held by firms that are not JTC1 members if they cited the SEPs or were cited by the SEPs. Thus, after deleting the biased citations in which the year cited is later than the year citing, we obtained more than 15,000 pairs of patent citations that were cited during the period of 1990–2010 in our sample.

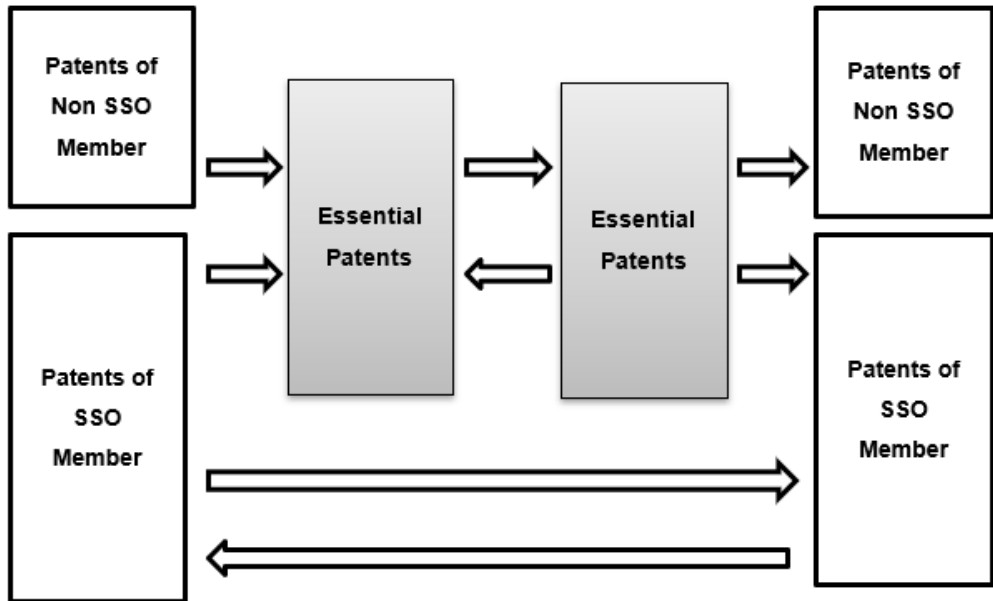

**Figure 1.** Patent citation relationships for the sample.

*2.3. Main-Path Analysis in Social Network Analysis*

The developments in the field of social network analysis have created several software tools that can facilitate the visualization, analysis, and interpretation of cooperation and citation networks, such as R, Pajek, Gephi, etc,. At the same time, these software tools can also explain the relationships between technology fields, patent applications, inventors, applicants, etc. Furthermore, the tools facilitate and support the combination of the qualitative and quantitative analyses of social networks, allowing for the construction of an index to explore the characteristics of the network for the applications on knowledge learning, knowledge flows, transfers, etc. [16,18–22].

In general, an item receiving more citations is deemed to be of more importance. In most citation networks, however, all patents are linked in one bicomponent. This cohesion concept does not take time into account. It does not reflect the incremental development of knowledge nor does it identify the patents that were vital to this development. Therefore, a special technique for citation analysis was developed that explicitly focuses on the flow of time. It is called "main-path" analysis [12].

Let us think of a citation network as a system of channels that transports scientific knowledge or information. A patent that integrates information from several previous items and adds substantial new knowledge receives many citations, and it will render the citations to previous patents more or less redundant. As a consequence, it is an important junction of channels and a great deal of knowledge flows through it. If knowledge flows through citations, a citation that is needed in the paths between many patents is more crucial than a patent that is hardly needed for linking patents. The most important citations constitute one or more main-paths, which are likely to be the backbones of a technology tradition.

Main-path analysis calculates the extent to which a particular citation or patent is needed for linking patents, which is called the traversal count, or traversal weight, of a citation or a patent. First, the procedure counts all paths from each source (a patent that is not citing within the dataset) to each sink (a patent that is not cited within the dataset), and it counts the number of paths that use a citation by the total number of paths between the sources and sink vertices in the network. This proportion is the traversal weight of a citation. In this paper, we employ an algorithm called the search path link count (SPLC), that weights each edge proportionally to how often a given link is present in all the paths that can link between any start point (i.e., patents that do not cite any other patent) to any

endpoint or sink (i.e., patents that do not receive any citation). Thus, the paths with the highest SPLC values are more likely to be on the main-path.

Bekkers and Martinelli (2012) [9] assume that the main-path is an accurate description of the most important contributions to the field, and one might expect that most of the patents on this main-path are indeed claimed to be essential to the standard.

### 2.4. Brokerage Roles in Social Network Analysis

The approach to brokerages and affiliations may help us to better understand the roles of patents that dominate a transactional, or exchange of knowledge, network. The roles of the actors in the network can be quite divergent and are categorized as "itinerant", "representative", "gatekeeper", and "liaison". The approach of the knowledge brokerage analysis has been identified as a promising strategy for investigating knowledge positions [18–21,23,24].

Group affiliation is often important in the brokerage process [23,24]. In real-world patent citation relationships, patents tend to be cited within the same field, and the patent that mediates between different fields plays the brokerage role. We may imagine that a patent, B, in the manufacturing field is cited to another patent in the IT field; for the knowledge transfer path, patent B is important because whoever owns B has a strong negotiating position and the chance to strike a good deal, and the removal of B disconnects the knowledge transfer path. In our paper, the patents in the same field, or that are often cited mutually, are comprehended as a group, and we wanted to find out if patents act as brokerage roles in our sample.

Figures 2–6 depict the categories, where the triad in which actor B mediates the transactions between actor A and actor C can display five different patterns of field affiliations:

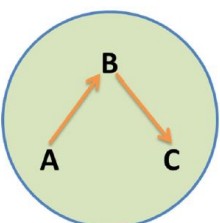

**Figure 2.** Coordinator.

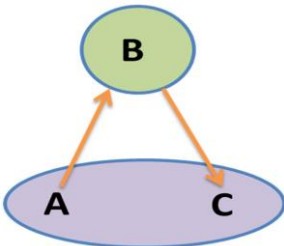

**Figure 3.** Itinerant.

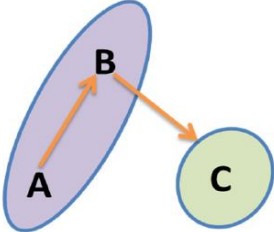

**Figure 4.** Representative.

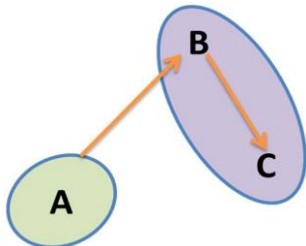

**Figure 5.** Gatekeeper.

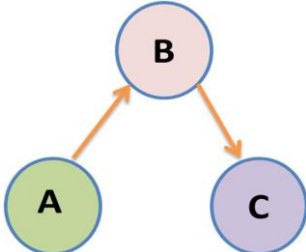

**Figure 6.** Liaison.

- In the "coordinator" triad, all actors, including broker B, and the source of knowledge, are in the same group.
- In the "itinerant" framework, broker B mediates between the actors, A and C, that are in the same group, but broker B is not part of this group.
- The "representative" role is given if a cluster delegates the role of brokering external knowledge to someone in the other group.
- The "gatekeeper" screens external knowledge and distributes it within its own group.
- "Liaison" refers to when the knowledge is brokered across different groups, none of which broker B is a member of.

*2.5. Other Covariants*

We acquired data related to the determinants of the strategies of the SSO member firms on the SEPs from the Searle Center Database [3]. The data comprises the number of employees, the number of patent applications, and the ratio of R&D expenditure to total sales for the JTC1 member firms in the sample period. With regard to the timing for cooperation and entry into the JTC1, we employed the year of the first pool launch for the JTC1 that was also released in the Searle Center Database. All statistical descriptions are shown in Table 1.

**Table 1.** Statistical descriptions (1).

| Variable | Obs | Mean | Std. Dev. | Min | Max |
|---|---|---|---|---|---|
| *No. of Employees* | 2419 | 80,416 | 70,646 | 1 | 264,880 |
| *Sales (MM US$)* | 2270 | 25,300 | 17,987 | 1 | 71,186 |
| *No. of Patent Applications* | 2229 | 2692 | 3459 | 2 | 11,424 |
| *R&D Expenditure (MM US$)* | 2254 | 1952 | 1369 | 1 | 3872 |
| *Year of First Pool Launch* | 2392 | 1995 | 5 | 1990 | 2005 |

## 3. Social Network Analysis

This section highlights some characteristics of patents in the JTC1 by employing methodologies currently developed in practice. These types of network analyses allow for the identification of the important players in the JTC1, and their connectedness, which can be used in the analyses of competitors or partners in this SSO.

*3.1. Investigating the Presence of Essential Patents on the Main-Path*

To implement the network analysis, we used Pajek 5.0.9, a software tool for analyzing social networks [24], to measure the SPLC values for the main-path in the patent citation network. Figure 7 illustrates a selected citation network in which the values (or weight) of the SPLC were larger than 0.004. The network consists of 180 patents, and out of those, the 41 patents with the sky-blue marks are the SEPs. Figure 7 also shows, in a solid line, the main-path with the highest SPLC values. The main-path comprises a total of eighteen patents, of which five patents are essential patents, and thirteen patents are other patents that are not claimed by the SSO member firms.

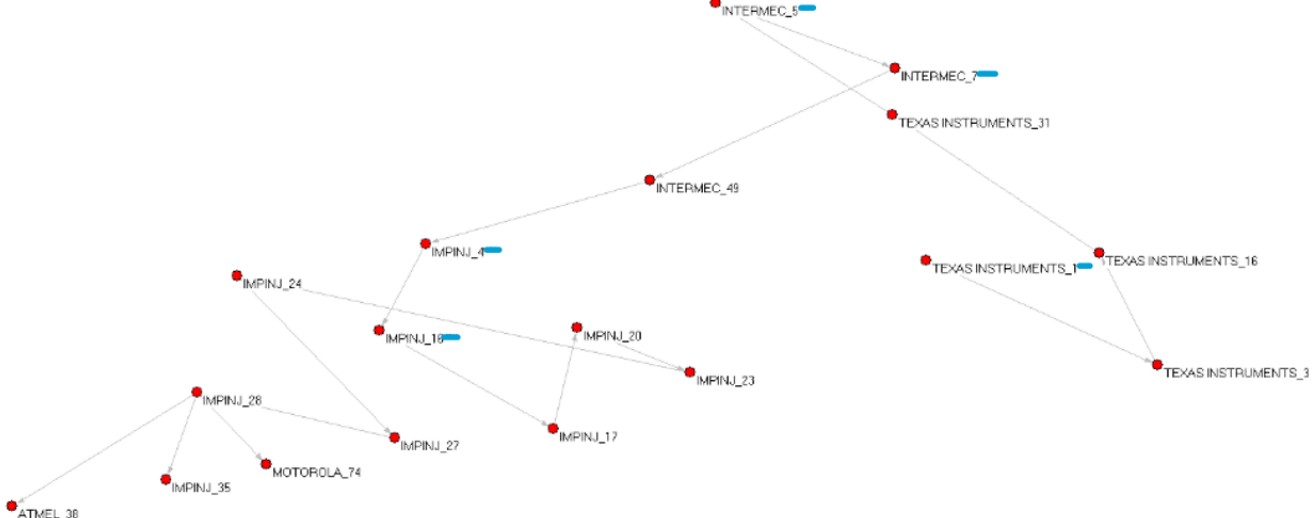

**Figure 7.** Main-path in selected citation network. Note: patents with sky-blue marks denote essential patents claimed by their owners.

Furthermore, Table 2 and Figure 8 reveal the distributions of the SPLC values for the SEPs, as well as the patents that are not claimed in the selected network. Although the average values of the SPLC for the SEPs are larger than those for non-claimed patents, compared with the latter, the former does not overwhelmingly contribute to the main-path. Thus, our finding in the JTC1 is consistent with that in Bekkers and Martinelli (2012) [9].

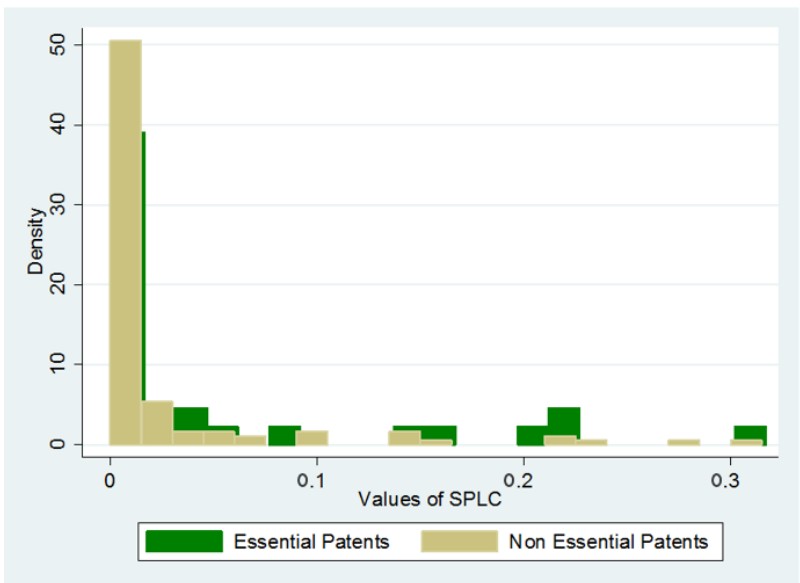

**Figure 8.** Histogram of the values of SPLC in selected network.

**Table 2.** Quantile of the values of SPLC in selected network.

| | Values of SPLC | | | | | | | | | |
|---|---|---|---|---|---|---|---|---|---|---|
| | **Mean** | **1%** | **5%** | **10%** | **25%** | **50%** | **75%** | **90%** | **95%** | **99%** |
| **Non Essential Patents** | 0.00135 | 0.00000 | 0.00000 | 0.00000 | 0.00001 | 0.00002 | 0.00009 | 0.00059 | 0.00179 | 0.01812 |
| **Essential Patents** | 0.00695 | 0.00000 | 0.00000 | 0.00000 | 0.00002 | 0.00006 | 0.00046 | 0.00334 | 0.01258 | 0.21333 |

### 3.2. Visualization Analysis on Brokerage Roles

While the main-path approach is widely used, and results in a valid representation of the main-path of technological development, such an approach is likely to face serious limitations. The question is whether such an "over selective" path lacks the necessary degree of granularity. Some companies might have contributed important knowledge, but their patents are not part of the main-path themselves. Recognizing these restrictions, this paper proposes an alternative approach that makes it more apt to evaluate knowledge positions, that is, the brokerage-role approach.

Social ties are one measure of social capital that can be used by actors for positive advantages. If we pay attention to the ties between an actor's contacts, an actor that is connected to actors who are themselves not directly connected has opportunities to mediate between them, and to profit from this mediation [23].

Research into brokerage roles is concerned with describing the types of brokerage roles that dominate a transactional or exchange network. In addition, individual positions within the network may be characterized by the dominant type of brokerage role, and hypotheses may be tested about the personal characteristics of individuals with certain types of brokerage roles.

Figures 9–12 demonstrate the values of "itinerant", "representative", "gatekeeper" and "liaison" for the selected network, which were measured by the Pajek. The size of the nodes shows the extent to which the patents play a particular brokerage role in the patent citation network. Because the values of "coordinator" are highly correlated with those of "itinerant", we discarded them from our analysis. The number of brokers in full network, and statistical descriptions for those brokers are summarized in Tables 3 and 4.

The size of the nodes counts the frequency of broker roles in our network. For example, patent no. 5, from INTERMEC Co., plays "itinerant" and "liaison" many times, but has no role as "representative" or "gatekeeper", so it has a large size in Figures 9 and 12, but a tiny size in Figures 10 and 11.

Figure 9 shows that, among the ten patents with the largest values of "itinerant", only no. 5 and 49 of the patents from INTERMEC Co. are located on the main-path, which is described by a solid line, while the other eight patents (no. 6, 41, and 49 from INTETMEC; no. 10 from SYMBOL TECH; no. 1 from MATRIC; no. 1 from ZEBRA TECH; no. 4 from ALIEN TECH; no. 14 from TEXAS INS; and no. 1 from BTG INT) are far from the main-path. Furthermore, out of the ten patents, only two are nonessential patents. Thus, the brokerage approach provides us with more information for understanding the roles of the SEPs, compared with the main-path approach.

At the same time, as can be seen from Table 3, among the 241 SEPs, approximately 76 and 81% of the SEPs play the roles of "itinerant" and "liaison", respectively, while those of the patents not claimed are less than 2%. Our findings suggest that there is a strong relationship between the broker roles, such as "itinerant" and "liaison", and the SEPs, which means that the patents that serve as "itinerant" and "liaison" may be more likely to be claimed as SEPs. On the other hand, however, only 2.9% of the essential patents are "representative", and 1.66% of the essential patents are "gatekeepers", which are not significantly different than the 1.98 and 1.12% for the patents not claimed.

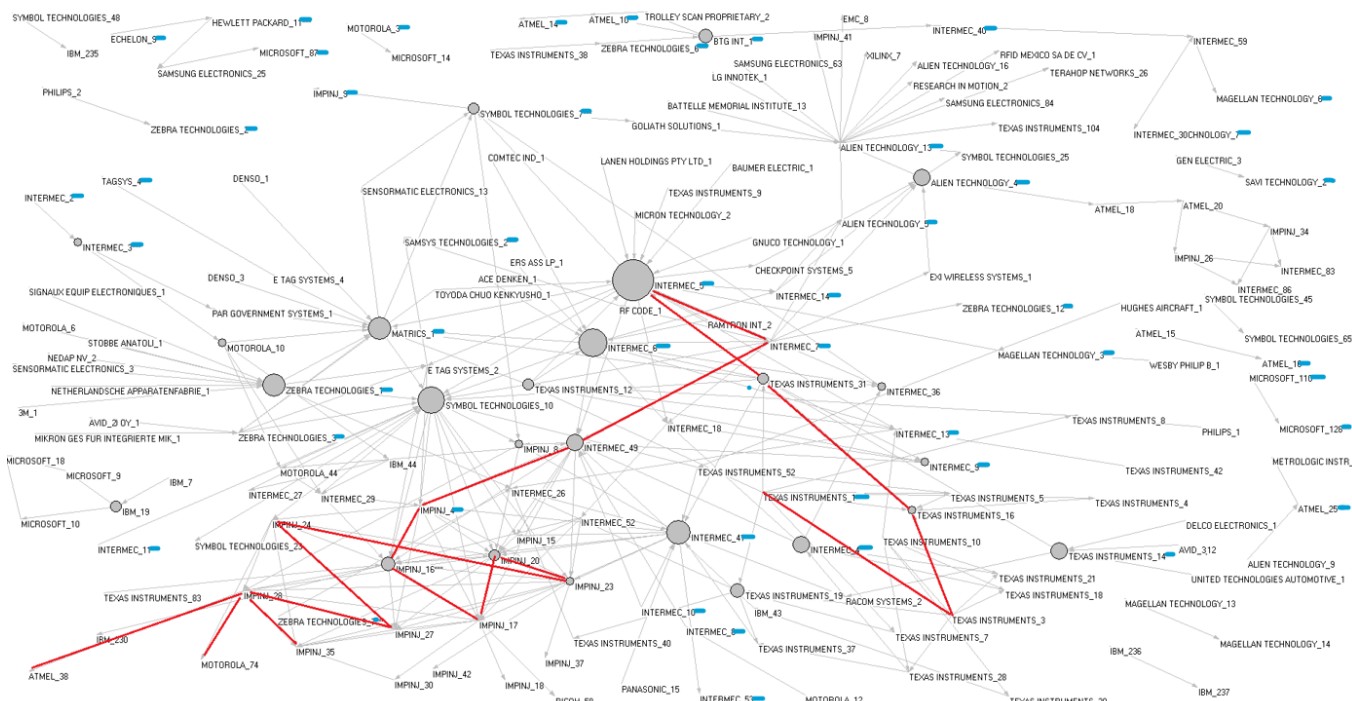

**Figure 9.** Values of "itinerant" in selected network.

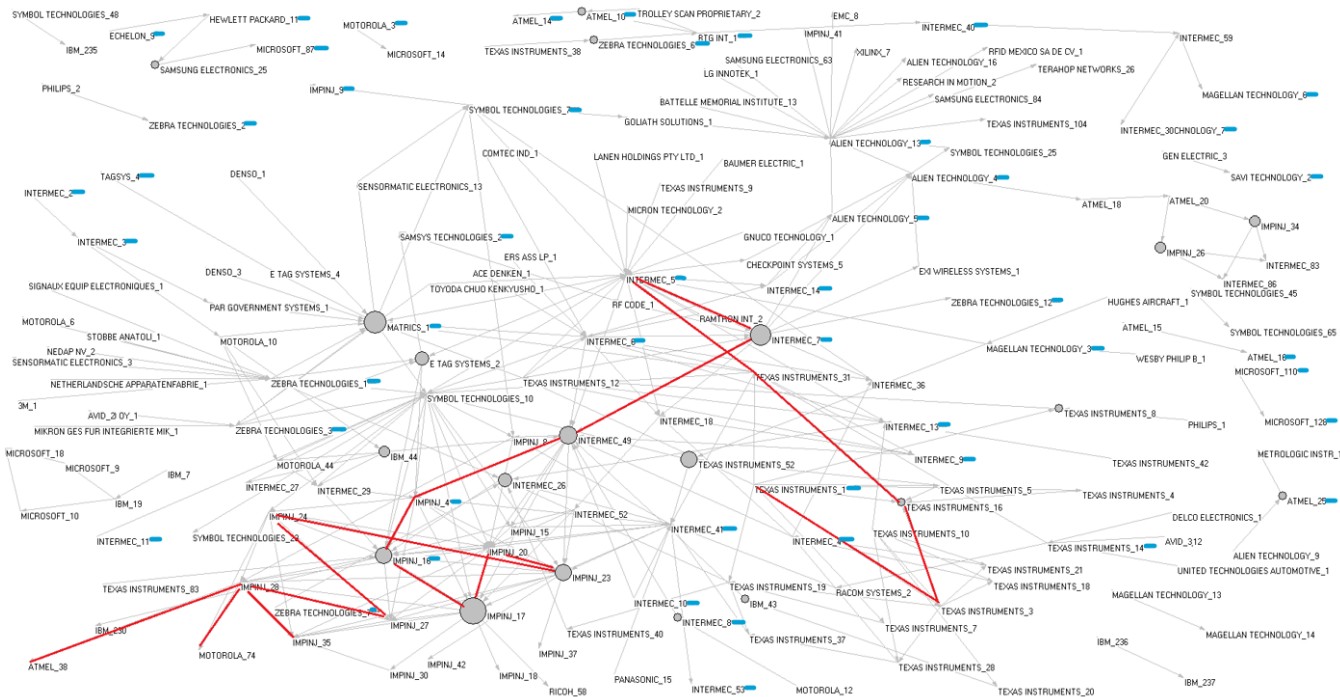

**Figure 10.** Values of "representative" in selected network.

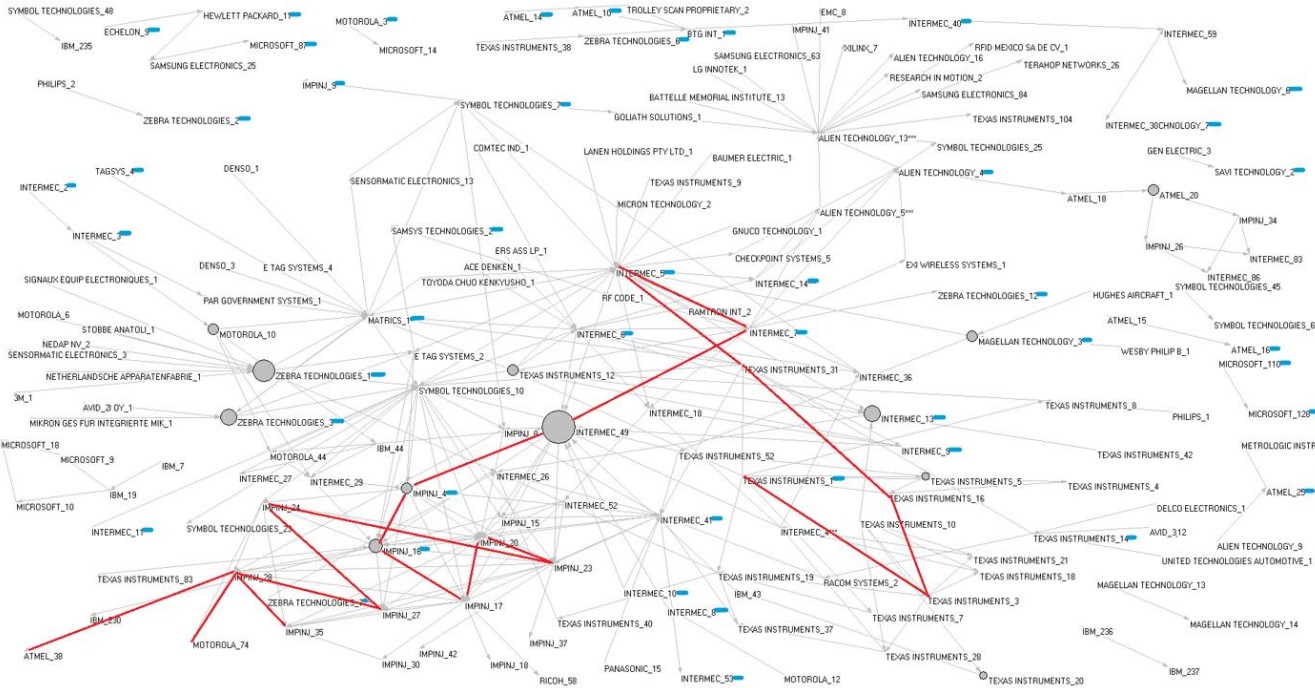

**Figure 11.** Values of "gatekeeper" in selected network.

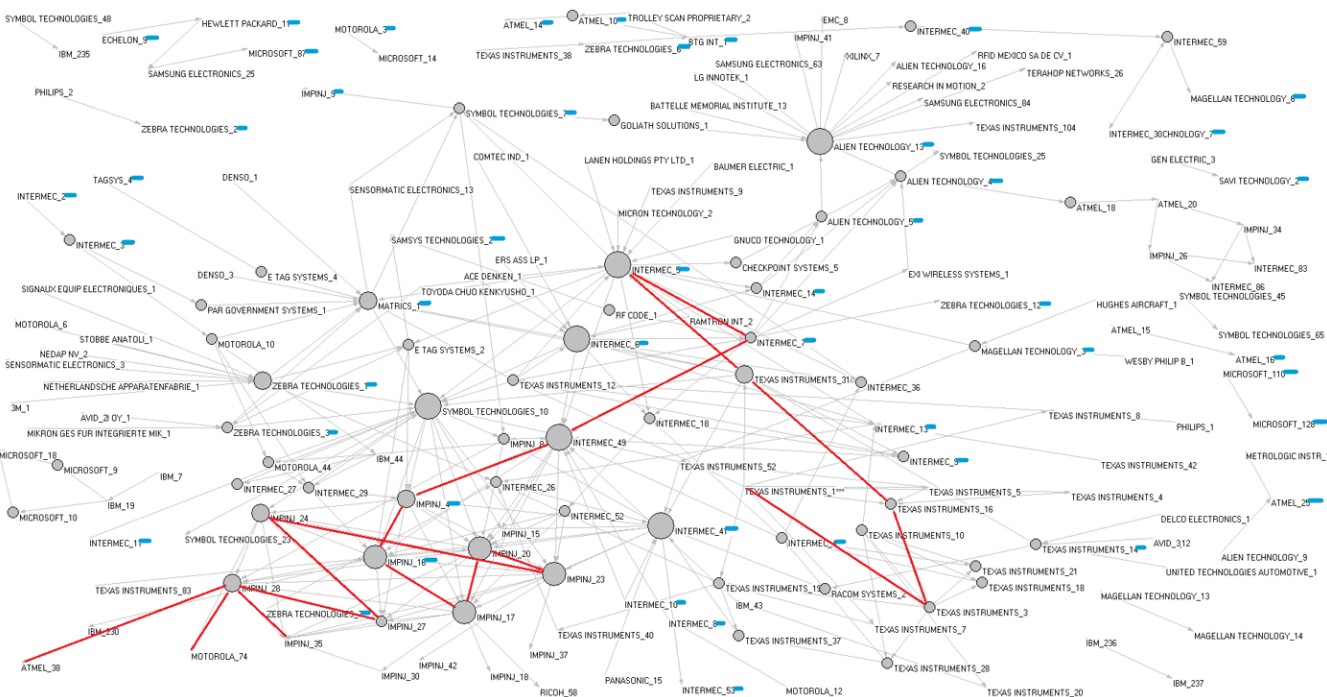

**Figure 12.** Values of "liaison" in selected network. Note: patents with sky-blue marks denote essential patents claimed by their owners in Figures 9–12.

Our social network analysis provides evidence that we are confronted with a selection effect: the values of essential patents are not only more strongly cumulative, but also more valuable than the nonessential patents from their technological field. This could result from the fact that the SSOs often choose between different technological options and select the best technologies for inclusion into the standard. We also find that the SPLC values of the essential patents are not absolutely larger than those of the nonessential patents, which implies that the patents claimed to be essential are not necessarily on the main-path.

**Table 3.** Number of brokers in full network.

| | Number of Brokers | | |
|---|---|---|---|
| **Roles of Brokers** | **Nonessential Patents** | **Essential Patents** | **Total** |
| *Itinerant* | 49 (1.90%) | 183 (75.93%) | 232 (100.00%) |
| *Liaison* | 338 (1.12%) | 194 (80.50%) | 532 (100.00%) |
| *Representative* | 29 (1.12%) | 7(2.90%) | 36 (100.00%) |
| *Gatekeeper* | 51 (1.98%) | 4(1.66%) | 55 (100.00%) |
| **Total** | 2581 | 241 | |

**Table 4.** Statistical descriptions (2).

| **Variable** | **Obs** | **Mean** | **Std. Dev.** | **Min** | **Max** |
|---|---|---|---|---|---|
| *Dummy for Essential Patents* | 2822 | 0.085 | 0.280 | 0 | 1.000 |
| *Values of SPLC* | 2822 | 0.002 | 0.016 | 0 | 0.311 |
| *Betweenness Centrality* | 2822 | 0.002 | 0.011 | 0 | 0.218 |
| *Itinerant* | 2822 | 4.914 | 44.638 | 0 | 1220.000 |
| *Liaison* | 2822 | 22.427 | 176.095 | 0 | 4529.000 |
| *Representative* | 2822 | 0.053 | 0.720 | 0 | 22.000 |
| *Gatekeeper* | 2822 | 0.047 | 0.471 | 0 | 14.000 |

## 4. Empirical Analysis on the Relation between Main-Path, Brokerage Roles, and SEPs

The aim of this section is to empirically explore the relations between the main-path, brokerage roles, and the SEPs, and investigate the determinants of the SEPs.

We build a dependent variable related to the SEPs, where it equals the unit if the patent is claimed to be essential, and it is zero otherwise.

We also consider the regressions with the index for "betweenness centrality", which calculate the extent to which an actor (or a patent) is located on the shorted path between any two nodes in the patent citation network. Betweenness centrality captures both the centrality and the spanning of structural holes in the network [25]. For an actor (or patent), *i*, its value of betweenness centrality can be measured by:

$$Betweenness\ Centrality_i\ =\ \sum_{j \neq k \neq i} \frac{g_{jk}(i)}{g_{jk}}$$

where $g_{jk}(i)$ denotes the number of shortest paths linking actors, *j* and *k*, that contain the focal actor, *I* and $g_{jk}$ is the total number of shortest paths from actor *j* to actor *k*.

### 4.1. Baseline Regressions

Table 5 presents the results of the regression analyses for the impacts of the SPLC value and brokerage roles on the patents claimed as SEPs. First of all, the coefficients for the "itinerant" are strongly positive and significant in both of our two regression models, indicating that the patents with the itinerant position are more likely to be claimed essential.

The liaison is not significant against the null, which may be due to the multicollinearity problem (see the correlation coefficients in Table 5). We tested the variance inflated factor (VIF) for all the independent variables in Table 6. The VIF is up to 3.83 for the "liaison", relative to 3.44 for the betweenness centrality, 1.55 for the "itinerant", and 1.0 for the three other variables. Although there is a debate about the critical values of the VIF, according to Hair et al., (2010) [26], if the VIF value is around, or exceeding, 4.0, then a problem with multicollinearity should be considered. Thus, we excluded the "liaison" in the second regression model, and the "itinerant" stayed the same.

At the same time, the coefficients for the "representative" were also positive and significant. However, the significance level seemed to be weak. In contrast to those for the "itinerant" and "representative", however, we can find that the coefficients for the SPLC

value are not significant in either of our models, which again verified our conclusion that patents on the main-path are not necessarily essential patents.

**Table 5.** Correlation coefficients of variables.

| | *1* | *2* | *3* | *4* | *5* | *6* | *7* | *8* | *9* | *10* | *11* | *12* |
|---|---|---|---|---|---|---|---|---|---|---|---|---|
| **1. Dummy for Essential Patents** | 1.000 | | | | | | | | | | | |
| **2. Values of SPLC** | 0.114 | 1.000 | | | | | | | | | | |
| **3. Betweenness Centrality** | 0.297 | 0.054 | 1.000 | | | | | | | | | |
| **4. Itinerant** | 0.376 | 0.036 | 0.461 | 1.000 | | | | | | | | |
| **5. Liaison** | 0.325 | 0.022 | 0.728 | 0.534 | 1.000 | | | | | | | |
| **6. Representative** | 0.019 | −0.006 | 0.020 | −0.006 | 0.024 | 1.000 | | | | | | |
| **7. Gatekeeper** | −0.018 | 0.023 | 0.050 | −0.008 | 0.004 | −0.008 | 1.000 | | | | | |
| **8. No. of Employees** | −0.051 | −0.019 | −0.071 | −0.051 | −0.053 | −0.035 | −0.037 | 1.000 | | | | |
| **9. Sales** | −0.025 | −0.040 | −0.074 | −0.031 | −0.023 | −0.020 | −0.058 | 0.917 | 1.000 | | | |
| **10. No. of Patent Applications** | −0.113 | −0.025 | −0.072 | −0.056 | −0.052 | −0.025 | −0.062 | 0.252 | 0.237 | 1.000 | | |
| **11. R&D Expenditure** | −0.014 | −0.051 | −0.001 | 0.008 | 0.067 | 0.040 | −0.023 | 0.304 | 0.550 | −0.011 | 1.000 | |
| **12. Year of First Pool Launch** | 0.093 | 0.085 | 0.096 | 0.033 | 0.017 | 0.028 | 0.097 | −0.496 | −0.592 | −0.406 | −0.379 | 1.000 |

Furthermore, the coefficient for the betweenness centrality is not significant, indicating that high betweenness centrality values are not contributing to a firm's patents being claimed essential.

**Table 6.** Baseline estimations with logit regression model.

| Covariables | I | II |
|---|---|---|
| Dependent Var: | | |
| *Dummy for SEPs* | | |
| *Values of SPLC* | 5.286 | 5.413 |
| | (1.11) | (1.12) |
| *Itinerant* | 0.656 *** | 0.705 *** |
| | (4.65) | (4.56) |
| *Liaison* | 0.005 | |
| | (0.80) | |
| *Representative* | 0.152 * | 0.159 ** |
| | (1.74) | (2.09) |
| *Gatekeeper* | 0.152 | 0.147 |
| | (1.09) | (1.03) |
| *Betweenness Centrality* | | 16.035 |
| | | (1.48) |
| Log Likelihood | −232.77 | −233.30 |
| No. of Obs. | 1819 | 1819 |

Note: (1) All regressions include fixed effects for the SSO member firms. (2) The values in the parenthesis are t statistics. (3) "***", "**", and "*" denote significance levels at 1, 5, 10% respectively.

### 4.2. Estimates for Determinants of Strategies Related to the SEPs

Table 7 allows us to underline a couple of the results. First, we can obtain the same conclusion as Table 6: that a strong link exists between the declaration to be essential and the patents serving as "itinerant" and "representative". Moreover, the coefficients for the SPLC and the betweenness centrality allow for the refinement of the previous results from the last table. Furthermore, the firms in the "gatekeeper" position do not seem to help their patents to be claimed essential.

With regard to the determinants of the strategies of the SSO member firms, the estimates of the number of patent applications and the R&D intensity are revealed to be positive and significant in some cases, suggesting that the SSO member firms with larger

patent portfolios, and that engage in more R&D activities, are more likely to claim their patents to be essential. However, the impacts of the number of employees are mixed.

**Table 7.** Logit estimates for determinants of strategies related to SEPs.

| | I | II | III | IV | V | VI | VII | VIII |
|---|---|---|---|---|---|---|---|---|
| Dependent Var: *Dummy for SEPs* | | | | | | | | |
| *Values of SPLC* | 7.455 | 5.582 | 7.237 | 7.237 | 7.183 | 5.534 | 7.119 | 7.119 |
| | (1.35) | (1.12) | (1.31) | (1.31) | (1.29) | (1.12) | (1.29) | (1.29) |
| *Itinerant* | 0.653 *** | 0.671 *** | 0.658 *** | 0.658 *** | 0.559 *** | 0.640 *** | 0.562 *** | 0.562 *** |
| | (4.47) | (5.68) | (4.54) | (4.54) | (5.34) | (4.89) | (5.33) | (5.33) |
| *Liaison* | −0.006 | −0.002 | −0.006 | −0.006 | | | | |
| | (−0.75) | (−0.29) | (−0.78) | (−0.78) | | | | |
| *Representative* | 0.149 ** | 0.159 ** | 0.149 ** | 0.149 ** | 0.146 ** | 0.158 ** | 0.146 ** | 0.146 ** |
| | (2.28) | (2.28) | (2.27) | (2.27) | (2.20) | (2.25) | (2.20) | (2.20) |
| *Gatekeeper* | −0.473 | −0.274 ** | −0.478 | −0.478 | −0.482 | −0.273 ** | −0.485 | −0.485 |
| | (−0.74) | (−2.32) | (−0.75) | (−0.75) | (−0.77) | (−2.35) | (−0.77) | (−0.77) |
| *Betweenness Centrality* | | | | | 5.508 | 10.438 | 4.018 | 4.024 |
| | | | | | (0.31) | (0.90) | (0.21) | (0.21) |
| *Log of Employees* | 1.215 *** | −2.161 *** | | 1.460 *** | 1.180 *** | −2.158 *** | | 1.252 *** |
| | (4.79) | (−3.95) | | (3.29) | (3.91) | (−4.14) | | (3.26) |
| *Log of Patent Applications* | 0.175 | | 1.514 *** | 0.304 | 0.183 | | 1.279 *** | 0.241 |
| | (0.70) | | (3.01) | (1.08) | (0.73) | | (2.98) | (0.90) |
| *Ratio of R&D to Sales* | 31.603 ** | | | | 29.487 * | | | |
| | (2.08) | | | | (1.78) | | | |
| *Year of First Pool Launch* | −0.006 | 0.372 *** | 0.461 *** | 0.197 * | −0.006 | 0.350 *** | 0.373 *** | 0.147 |
| | (−0.06) | (4.33) | (2.93) | (1.87) | (−0.06) | (3.77) | (2.74) | (1.48) |
| Log Likelihood | −168.46 | −186.76 | −168.38 | −168.38 | −168.96 | −186.70 | −168.91 | −168.91 |
| No. of Obs. | 1287 | 1409 | 1287 | 1287 | 1287 | 1409 | 1287 | 1287 |

Note: (1) All regressions include fixed effects for the SSO member firms. (2) The values in the parenthesis are t statistics. (3) "***", "**", and "*" denote significance levels at 1, 5, 10%, respectively.

What is noticeable is that the coefficients for the "year of first pool launch" are strongly positive and significant for our five models. We can infer from this result that the new patents launched in the pool are more likely to be claimed essential. This may be due to the fact that SSO member firms make contributions to standards under development. The later the firm first launches the SEPs to the pool, the more it claims the patents to SEPs.

## 5. Conclusions

In this paper, we implemented the empirical analysis of the knowledge positions of firms in high-tech standards-based markets by using the patent data at the firm level. We believe that being able to assess the knowledge positions is important because they are assumed to increase the chances for sustainable market participation, bargaining power, and licensing revenues. Our study focused on the JTC1, an SSO that provides a standards development environment related to the development of worldwide ICT standards for business and consumer applications. We attempted to utilize the social network technique to find out the characteristics, i.e., the "main-path", the "betweenness centrality", and the brokerage roles in the patent citation network of the JTC1, and carried out a regression analysis of the effects of these characteristics on the declaration of the SEPs. Our main conclusions are:

- Patent citations are used as a tool for mapping technological trajectories, and the main-path analysis has been widely used in previous research. The main-path analysis does identify the most important technological advances and breakthroughs in the development of this technology; however, it is too selective to fully assess the knowledge positions of firms.
- To fill this research gap, we investigated another social network analysis method: the brokerage-roles approach, which results in a better measurement of the knowledge positions, and more suitably matches the outcomes of the historical/technical narrative and analysis of knowledge flows. We also implemented an empirical analysis to support our findings based on the social network analysis, and empirical results

revealed that, the relationships between the brokerage roles and the SEPs are strongly significant in our sample, while it is not the case for the main-path values.

- While investigating the deep-seated reasons for the former two conclusions, we found that claims of essentiality are the result of the strategic behavior of the patent's owner. As important patents often occupy brokerage positions, we can infer from our results that firms usually tend to pursue their knowledge positions in the standard activities, or a competitive advantage in their international business, by claiming their really important patents to be essential in the standardization.

**Author Contributions:** Conceptualization, J.J. and X.Z.; methodology, J.J. and X.Z.; software, J.J.; validation, J.J. and X.Z.; formal analysis, J.J. and X.Z.; investigation, J.J. and X.Z.; resources, J.J.; data curation, J.J.; writing—original draft preparation, J.J. and X.Z.; writing—review and editing, J.J. and X.Z.; visualization, J.J. and X.Z. All authors have read and agreed to the published version of the manuscript.

**Funding:** This research received no external funding.

**Institutional Review Board Statement:** Not applicable.

**Informed Consent Statement:** Not applicable.

**Data Availability Statement:** We collected our data for standard essential patents from the Disclosed Standard Essential Patents (dSEP) Database developed by Bekkers et al., (2012). The data are available in a publicly accessible repository: http://ssopatents.org/ (accessed on 3 September 2021). We acquired patent citation data from the patent dataset, PATSTAT ver. October 2016. PATSTAT, also known as the EPO Worldwide Patent Statistical Database, is a snapshot of the EPO master documentation database (DOCDB), with worldwide coverage, covering more than 20 tables, with bibliographic data of about 70 million for the patents issued by most of the patent institutes in the world. See https://www.epo.org/index.html (accessed on 3 September 2021). We acquired data related to the determinants of the strategies of the SSO member firms on the SEPs from the Searle Center Database. The data are available in a publicly accessible repository: http://www.law.northwestern.edu/research-faculty/searlecenter/innovationeconomics/data/technologystandards/ (accessed on 3 September 2021).

**Conflicts of Interest:** The authors declare no conflict of interest.

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
