# Peer review of "Essential Patents and Knowledge Position, a Network Analysis on the Basis of Patent Citations"

_standards, doi:10.3390/standards1020009_

Round 1
Reviewer 1 Report
Comments and Suggestions for Authors
Thanks for the authors submitting an interesting manuscript. However, I identified several critical issues, which need to be solved before publication.
General Issues
- Check the citation style of Standards and update references: I recommend the authors revise references based on the Standards template available online.
- Please, send the manuscript for English proofreading and check typos before submitting the revised manuscript.
- Omission of citations: There are sentences without any citations. For me, it looks pretty unnatural to write a paragraph for a manuscript without any citations. Because there are many articles already out there, it cannot be true writing a paragraph explaining research methods without any citations.
- Please, use abbreviations after you define the abbreviations. In the manuscript, the authors sometimes use the spell-out versions even they already introduced abbreviations. Sometimes abbreviations were introduced twice in the manuscript (See lines 31, 88).
Introduction
- It seems that contents in Section 1. Introduction and Section 2. Materials and Methods are mixed. For example, lines 56-82 in Section 1 explain the research method, which is also described in Section 2. Therefore, I recommend the authors reconstruct 'Introduction' by separating research methods to Section 2.
- I think that the authors want to explain why the manuscript's subject is important by describing the results of extant studies. However, the authors only list the results of extant studies, and the research gap is not fully explained in the manuscript. Therefore, I recommend that the authors rewrite Section 1 to clearly state the research gap while briefly summarizing the results of extant studies.
- The authors need to define the meaning of the knowledge position, which makes the manuscript differentiated from other studies.
Analysis result
- In lines 255-262, the authors showed visualized analysis results regarding brokerage roles. I believe that there needs to be more explanation on details of the analysis. ex) how does the size of nodes explain the brokerage roles? What makes differences in the path in terms of whether which node becomes the biggest one depending on the types of the brokerage roles?
Reviewer 2 Report
Comments and Suggestions for Authors
Dear Author(s),
Thank you for the opportunity to review the manuscript entitled ‘Essential Patents and Knowledge Position, a Network Analysis on the Basis of Patent Citations’. The paper addresses an interesting topic the standard essential patents (SEP) and empirically investigates the knowledge position measured using two approaches: one from the literature namely the ‘main-path’ and the other proposed by the authors namely the brokerage process. The paper also studies the determinants of the SSO declaring a patent as a SEP or not. The paper overall is written well and the idea of using broker typology framework for measuring knowledge position in the context of SEP is novel and very good. The paper however lacks methodological details necessary for assessing the rigor in obtaining the results. Also, some of the results are contradictory and need to be explained clearly in the result section.
Below I provide my detailed comments which I hope will help improve the paper further towards publication.
[1] My first major concern is with the method section. The methodology of the paper is not sufficiently explained and lacks critical details needed to assess the methodological rigor. For example, the brokerage process is claimed to be the novelty of the paper and the idea of using this framework to measure knowledge position is very good. But methodologically it is not clear how the patents in the sample (represented as nodes in the network figures) are classified under different broker types. While explaining the broker types on page 7, there is mention of different clusters using different colors. How are these clusters defined in the context of the citation network to classify a particular node (or patent) as “itinerant”, “representative”, “gatekeeper”, and “liaison”? These details must be included in the method section.
[2] The sample is said to consist of patents during the period 1990-2010. What is the year range for citations? It is possible that patents in 1990 and later have citations from years prior to 1990 and the patents in 2010 and before have citations from years after 2010. What are the boundary years for citation data? This needs to be explained in the method section.
[3] Has the year in which the patent is declared as SEP taken into consideration while building the citation network in Fig. 1, and Figs. 9 to 12? If not, then explain why it was not considered and if yes, then explain how the SEP declaration date was considered in the analysis.
[4] My second major concern is with the seemingly contradictory results in sections 3 and 4. In section 3, specifically in Table 4 and the results described, “itinerant” and “liaison” emerge as having a strong relationship with SEPs. But in the results explained in section 4, “Liaison” is not significant in determining SEP but instead “representative” is significant. Why is this? Please explain and discuss the potential reasons and implications of these varied findings.
[5] The paper starts by mentioning the “knowledge positions of the SEPs…”, but later in the conclusion section, the authors conclude by saying they investigate the “… knowledge positions of the firms…”. Please clarify and make it consistent throughout the text whether the focus of the presented network analysis is the knowledge position of patents or firms.
[6] In Table 1, indicate the units for Sales and R&D Expenditure.
[7] In all the network figures (Fig. 1, and Figs. 9 to 12), the SEPs marked with *** can be color-coded into a different prominent color. This way it will be visually clearer to see and understand the patterns. At present, it is difficult to see the distribution of SEPs in the network.
[8] It is not clear, what does ‘PNLC’ denotes in Tables 5 and 6.
[9] Finally, the paper has several typos and needs to be proofread. For example, Table 4 uses the term ‘Blockers’ which I think should be ‘Brokers’. Page 4, line 168, ‘patens’ to be corrected as ‘patents’.
I hope the authors find the comments useful to improve the paper. All the best!
Round 2
Reviewer 1 Report
Comments and Suggestions for Authors
I appreciate the authors' efforts to improve the manuscript. Overall, the quality of the manuscript has significantly improved, and several issues have been solved.
However, I believe that the manuscript needs to be further improved. Therefore, I suggest the authors consider issues as below:
- The introduction has been improved quite a lot. However, it seems that there is still room for further improvement. I require authors to condense information of research methodology (lines 74-103) in the introduction. A detailed description of the research methodology in Section I needs to be moved to Section II to avoid repetition.
- Regarding 2.2. Patent Citations, I suggest authors provide more detailed information on how they matched dSEP data with patent identities in the PATSTAT. Please, explain what criteria were used during the matching process. For example, the authors explained that they used patent identities in PATSTAT to merge the dSEP with USPTO patent citation information. However, it does not sufficiently explain details of how authors manage this process. I believe that readers will want to know more about detailed information about the merge process.
- In Section III, the authors highlight features and developments of social network analysis. Since research methodology and data are closely related, providing information on data and methods in the same Section will reduce repetition. Therefore, I suggest authors move the methodological description on social network analysis to Section II and only provide analysis results in Section III.
- The authors suggested three conclusions in the manuscript. However, the way of presenting conclusions needs to be further improved. Currently, the authors' way of providing conclusions lacks a detailed discussion on why their findings are noble and meaningful. The authors can provide in-depth implications and discussions based on their research findings regarding methodological and strategic aspects.
Reviewer 2 Report
Comments and Suggestions for Authors
Dear Authors,
Good work with the revised and resubmitted manuscript which has improved in terms of its overall clarity in the methods and the results sections. You have addressed most of my review comments adequately, except for my comment [4] which needs further clarification and revision to the manuscript.
In your response to my comment [4], you mention that 'Actually, the liaison is not significant against the null in Table 6 and 7. This may be due to strong correlation between the liaison and itinerant, which result in somewhat multicollinearity problem'. If there is a multicollinearity problem then this has to be tested and adjusted for in the regression model. I suggest you test for multi-collinearity and if exists then report the multicollinearity adjusted regression results.
Otherwise, the manuscript is almost ready. All the best with the publication!
Round 3
Reviewer 1 Report
Comments and Suggestions for Authors
I appreciate the authors' hard work to improve the manuscript. Since the manuscript has been improved quite a lot, I believe this manuscript is qualified for publication in Standards. Congratulations!